# Feasibility of Optical Genome Mapping in Cytogenetic Diagnostics of Hematological Neoplasms: A New Way to Look at DNA

**DOI:** 10.3390/diagnostics13111841

**Published:** 2023-05-24

**Authors:** Nicoletta Coccaro, Luisa Anelli, Antonella Zagaria, Francesco Tarantini, Cosimo Cumbo, Giuseppina Tota, Crescenzio Francesco Minervini, Angela Minervini, Maria Rosa Conserva, Immacolata Redavid, Elisa Parciante, Maria Giovanna Macchia, Giorgina Specchia, Pellegrino Musto, Francesco Albano

**Affiliations:** 1Hematology and Stem Cell Transplantation Unit, Department of Precision and Regenerative Medicine and Ionian Area (DiMePRe-J), University of Bari “Aldo Moro”, 70124 Bari, Italy; nicoletta.coccaro@hotmail.it (N.C.); luisa.anelli@uniba.it (L.A.); antonellazagaria@hotmail.com (A.Z.); francesco.tarantini@uniba.it (F.T.); cosimo.cumbo@gmail.com (C.C.); giuseppina.tota@uniba.it (G.T.); eziominervini@gmail.com (C.F.M.); minervini.angela@gmail.com (A.M.); mariarosaconserva@gmail.com (M.R.C.); imma.redavid.ir@gmail.com (I.R.); macchiamaryjoe@gmail.com (M.G.M.); pellegrino.musto@uniba.it (P.M.); 2School of Medicine, University of Bari “Aldo Moro”, 70124 Bari, Italy; specchiagiorgina@gmail.com

**Keywords:** cytogenetics, hematologic neoplasms, optical genome mapping

## Abstract

Optical genome mapping (OGM) is a new genome-wide technology that can reveal both structural genomic variations (SVs) and copy number variations (CNVs) in a single assay. OGM was initially employed to perform genome assembly and genome research, but it is now more widely used to study chromosome aberrations in genetic disorders and in human cancer. One of the most useful OGM applications is in hematological malignancies, where chromosomal rearrangements are frequent and conventional cytogenetic analysis alone is insufficient, necessitating further confirmation using ancillary techniques such as fluorescence in situ hybridization, chromosomal microarrays, or multiple ligation-dependent probe amplification. The first studies tested OGM efficiency and sensitivity for SV and CNV detection, comparing heterogeneous groups of lymphoid and myeloid hematological sample data with those obtained using standard cytogenetic diagnostic tests. Most of the work based on this innovative technology was focused on myelodysplastic syndromes (MDSs), acute myeloid leukemia (AML), and acute lymphoblastic leukemia (ALL), whereas little attention was paid to chronic lymphocytic leukemia (CLL) or multiple myeloma (MM), and none was paid to lymphomas. The studies showed that OGM can now be considered as a highly reliable method, concordant with standard cytogenetic techniques but able to detect novel clinically significant SVs, thus allowing better patient classification, prognostic stratification, and therapeutic choices in hematological malignancies.

## 1. Introduction

Optical genome mapping (OGM) is a new genome-wide technology that is emerging in clinical genetics laboratories, replacing standard diagnostic techniques, such as chromosome binding analysis (CBA), fluorescence in situ hybridization (FISH), chromosomal microarrays (CMAs), or multiple ligation-dependent probe amplification (MLPA), as a means of revealing structural genomic variations (SVs) (translocations, insertions, inversions, deletions, duplications, etc.), copy number variations (CNVs), and whole-chromosome aneuploidies in a single assay [1,2]. Diagnostic laboratories are adopting OGMs as a tool for detecting genomic anomalies in constitutional disorders and cancers. OGM is based on DNA labeling rather than sequencing, and is considered to be a bridging technique between methods based on sequencing and on whole chromosome analysis. In this regard, a new definition of “next-generation cytogenetics” has been introduced [1,2,3].

The most common platform currently used to perform OGM analysis is the Saphyr system from Bionano Genomics (San Diego, CA, USA) [1,4,5,6]. Ultra-long high-molecular-weight (UHMW) genomic molecules (0.15–2.5 Mb) are fluorescence labeled at specific short nucleotide sequence sites; the labeling is carried out with the enzyme DLE-1 that targets a 6-mer DNA motif, achieving a label density of about 15 labels per 100 kb in the human genome, corresponding to a periodicity of approximately every 5 kbp [1,7]. Recently, Bionano has introduced a new method named direct labeling and staining (DLS), based on the direct fluorophore insertion in specific DNA motifs, avoiding nicking and the occurrence of double-stranded breaks [8]. This new method shows a 50x improvement in the labeling contiguity of molecules compared to the system based on the DLE-1 enzyme [8]. Labeled DNA is loaded onto a chip that linearizes the molecules, and the obtained genome-wide fluorescent pattern is then scanned and compared with a reference genome to identify structural variants (Figure 1). This analysis resembles conventional karyotyping, as changes in “banding” patterns are converted into structural variants, but OGM shows a 100X–20,000X higher resolution depending on the structural variant identified and the analysis tool employed [7]. An average coverage of 300X and 80–100X is generally achieved for somatic and constitutional analysis; genome-wide numerical and structural alterations can be detected in about one week, showing a much higher resolution than with other diagnostic techniques such as conventional cytogenetics, FISH, and CNV microarrays [7]. Two bioinformatic systems can be used: a rare variant pipeline (RVP) or a de novo assembly (Figure 1). They mainly differ in their capacity to reveal low-allele-frequency variants. The RVP is designed to identify variants at low allele frequencies, prevalent in heterogeneous samples such as cancers or samples with genetic mosaicism. The RVP compares target genomic molecules to the reference genome and can detect both SVs and CNVs; the SV calling tool can identify SVs of at least 5 kbp to tens of Mbp long (general cut-off > 100 kbp), and new DNA fusions showing a variant allele frequency (VAF) threshold of about 5%. In contrast, the CNV algorithm mainly identifies large aberrations (from 500 kbp up to aneuploidies, at a VAF of at least 10%), such as partial aneuploidies and terminal deletions, that are missed by the SV calling [3,7]. On the other hand, the de novo assembly arranges all of the labeled molecules in a de novo genome assembly for each chromosome, achieving low sensitivity for rare events (VAF of at least 15–25%) but detecting smaller SVs, of about 500 bp, as compared to RVP [5,7]. The Bionano analysis software allows users to view and manipulate maps and SVs, providing a graphical interface that does not demand specific bioinformatics skills [3]. Different data visualization methods exist: Circos plot, genome browser view, and whole-genome plot (Figure 1). The Circos plot provides a summary of detected variants, a variant allele fraction profile, chromosome number, cytobands, and, optionally, gene locations. The genome browser view is an interactive visualization tool to analyze the variants on a chromosome. The whole-genome view shows the genomic localizations of all of the chromosomes with a copy number, absence of heterozygosity (AOH)/loss in heterozygosity (LOH), and VAF in three separate plots.

## 2. Advantages of OGM Technique

OGM is a more rapid, less labor-intensive approach, and avoids the cascade of conventional diagnostic tests based on CBA, FISH, CMA, or MLPA, with a consequent reduction in time and costs (Table 1). OGM provides a complete assessment of global genomic alterations compared to any other conventional molecular or cytogenetic single test. Each of the different molecular and cytogenetic standard diagnostic techniques has limitations: conventional cytogenetics is based on the analysis of only 20 metaphases, can run into culture bias or artifacts, and shows a banding resolution of about 5 Mb [9,10,11,12,13,14,15]; FISH is characterized by a higher resolution but investigates few target genomic regions on the basis of karyotype suggestions [16,17,18,19,20,21,22,23], and CNV microarrays show a resolution of a few kb but cannot reveal balanced chromosomal rearrangements such as translocations or inversions, and cannot define the location of copy number gains [3,23,24,25,26,27,28,29,30]. In fact, the combination of SV and CNV calling via OGM highlights the fact that genomic losses and gains revealed using CNV microarrays can often be the consequence of an unbalanced translocation not detected by microarrays [3]. OGM can identify novel translocations leading to gene disruption or new fusions involving genes that can be important drivers of cancer pathogenesis and also of targeted therapy [3]; one such example is a novel balanced translocation involving *UBE3C* and *MSI2* fusion identified as a single chromosomal rearrangement in acute myeloid leukemia (AML) [3]. Moreover, OGM is particularly useful in defining rearrangements involving genes with multiple possible partners, such as *KMT2A*, *MECOM*, *ETV6*, *NUP98*, or *IGHV* in hematological malignancies; these rearrangements are usually investigated using FISH with two color gene specific break-apart probes, which are not, however, able to identify the partner gene [31,32]. In this respect, an example is the identification via OGM of a rare *NUP98::TNRC18* fusion gene in an AML patient [33].

Analysis using OGM is particularly successful in defining complex genomic rearrangements or identifying additional genomic material and marker chromosomes (Table 1) [3,34]. OGM is especially powerful in studying chromoanagenesis, a complex phenomenon caused by many genomic alterations involving a few chromosomal regions [34]. Chromoanagenesis includes chromothripsis and chromoplexy, the first consisting of hundreds of genomic rearrangements caused by chromosomal shattering and random reassembly, and the second consisting of multiple chained translocations involving different chromosomes in a single catastrophic event [35,36,37].

OGM technology can reveal the occurrence of insertions and deletions as small as 500 bp via the de novo assembly pipeline and can define translocation breakpoints with a precision of a few kilobases; this resolution is higher compared to conventional analyses such as karyotyping, FISH, and CNV microarrays [1,3,4]. Moreover, compared to other techniques based on short- and long-read sequencing systems, OGM shows superior detection of structural variants because long DNA molecules, ranging from 500 bp to 1 M bp, are processed [7].

Another advantage of OGM is that, compared to CBA and FISH techniques, it does not require culture cells and is not subject to culture bias. Moreover, unlike other molecular technologies such as next-generation sequencing (NGS), there is no amplification step; this limits the risk of variant overestimation or underestimation due to PCR bias, and so it offers a more accurate evaluation of aberration frequency. Moreover, compared to whole-genome sequencing (WGS), OGM shows a greater medium coverage (about 200–300X against 60X) and is able to detect alterations with a low VAF, of about 5% [38]. OGM cannot replace NGS, although it can be considered to be a helpful complementary technique for cytogenetic aberration identification, because point mutations (SNV) and small SVs can be revealed only by NGS [3]. Therefore, NGS complemented by OGM can be considered as the perfect technique for complete and exhaustive genomic analysis (Figure 2).

## 3. Limitations of OGM Technique

Despite many advantages, the OGM technique has some limitations (Table 1). OGM data still need to be supported by CBA or CMA to identify and define ploidy changes, when these changes affect the entire chromosome set rather than the gain or loss in single chromosomes [1]. This misinterpretation is particularly important in hematological malignancies, especially for the diagnosis and prognostic stratification of acute lymphoblastic leukemia (ALL) or multiple myeloma (MM) cases, where hyperdiploidy and hypodiploidy frequently occur, correlated to a favorable or very poor prognosis, respectively. Triploidy can now be detected in constitutional analysis via a de novo assembly pipeline, whereas the detection of somatic ploidy changes is still being developed [7].

Another limitation is the occurrence of copy number neutral loss in heterozygosity (CN-LOH), which can currently be revealed only by a de novo assembly pipeline, where these regions show an evident decrease in heterozygous SV calls compared to those detected in controls [3,39].

Another important OGM limitation is that it fails to detect some sub-clonal anomalies present in a small sub-set of cells, but which can instead be detected via FISH or CBA analysis (Table 1) [40]. Comparing OGM and FISH data, it has been demonstrated that OGM can reveal abnormalities in at least 10–15% of cells [3,40]. Even if CBA has the advantage of analyzing at the single-cell level and can provide information on clonal architecture, it detects abnormalities only present in proliferating cells; therefore, the aberration incidence or the clone size is further influenced by the metaphase selection. Moreover, cytogenetic sensitivity for detecting small cellular clones is very low, as only 20 metaphases are usually analyzed. This limit is crucial in both prenatal and cancer diagnoses, where neoplastic sub-clones or mosaicism could be present.

In cases of false negative OGM results, it is sometimes necessary to lower the CNV filter threshold, setting it at about 5 Mb, or even to remove filtering to detect some alterations already identified using standard diagnostic techniques and achieve full concordance [3]. Therefore, a CNV algorithm improvement would be desirable in future versions.

Other OGM limitations are linked to breakpoint localization. OGM may miss the detection of some structural variants if the breakpoints are located within repeated regions or sequences not covered by OGM, such as centromeres or telomeres [3,7,40]. In fact, balanced whole arm rearrangements such as Robertsonian translocations or end-to-end telomere fusions cannot be revealed, and isodicentric chromosomes can be misinterpreted. Modifying the masking filter setting is also recommended to visualize events in masked regions. Furthermore, previous studies have demonstrated that abnormalities involving the pseudoautosomal region PAR1 are not detected via RVP analysis due to the similarity of the sequences on the X and Y chromosomes, but can be revealed using a de novo assembly pipeline [7,40,41,42].

Some studies have reported that OGM could yield false positive results, such as the identification of fusions, representing regions randomly showing the same labeling pattern; in this respect, one example is the *DUX4::FRG2B* rearrangement identified in a study by Rack et al. that was not further confirmed via FISH and RNA-Seq [40]. In some instances, although confirmed at the genomic level, fusion genes were not revealed via further expression analysis. Another example was reported by Neveling et al. and described some false positive SVs identified via OGM and not confirmed via CNV microarrays. In these cases, the breakpoints mapped in low-coverage genome sites or in “DLE-1 mask regions” were highly variable regions, including centromeres and telomeres with an unusually high CNV occurrence in normal individuals, and should therefore be filtered out via RVP analysis [3].

## 4. Application of OGM in Hematological Malignancies

OGM was initially employed to perform a genome assembly of microorganisms, such as yeast and bacteria, and then for genome research on plants and vertebrates [4,43]. This novel technique is now more widely used to study chromosome aberrations in prenatal or constitutional diseases and human cancer [2,3,44,45]. However, OGM has not yet achieved widespread application in solid neoplasms, mainly due to the difficulty in obtaining high-quality UHMW genomic DNA from cancer cells.

One of the most useful OGM applications is in hematological diseases such as acute leukemia or MM, where complex karyotypes are frequent and conventional cytogenetic analysis alone is insufficient, demanding further confirmation via ancillary techniques [3,7]. OGM is performed in a single experiment and can replace the combination of classical molecular and cytogenetic approaches, such as karyotyping, FISH, and microarrays as means of identifying all somatic SVs and CNVs characterizing hematological samples; it is useful for both diagnosis and defining the prognosis [3,7]. The most helpful pipeline for OGM analysis in hematological malignancies is RVP. However, for specific cases, such as B-ALL with *IGH-CRLF2* fusion and the involvement of X and Y chromosome pseudoautosomal regions, a de novo assembly can also be employed which helps to discriminate which of the sex chromosomes is rearranged [7].

The first studies tested OGM efficiency and sensitivity for SV and CNV detection, analyzing heterogeneous lymphoid and myeloid hematological sample groups compared with data obtained from standard cytogenetic diagnostic tests [3,46]. OGM precisely defines the genomic chromosome structure, mainly in cases of complex rearrangements, and provides a complete definition of global genomic alterations as compared to other single assays [3]. The superiority of OGM is evident in defining the origin of marker chromosomes, complex translocations, and chromoanagenesis [3]. In these general studies, a few CNVs > 5 Mb were revealed per sample, whereas an average of hundreds of SVs was detected, including all rearrangements: insertions, deletions, inversions, duplications, and inter- and intra-chromosomal translocations [3]. Complete concordance between OGM and standard techniques was reached for known aberrations in simple cases, bearing < 5 genomic alterations, and all of the more frequent chromosomal rearrangements recurrent in ALL were detected and confirmed via OGM analysis [3]. A small study was performed by Podvin et al. on two cases with hypereosinophilia associated with myeloid neoplasms and *PDGFRB* rearrangements. OGM allowed for the identification of rare partner genes involved in *PDGFRB* fusions: *PCM1* and *GOLGA4* mapping in 8p22 and 3p21, respectively. OGM highlighted the complex origin of chromosomal rearrangements bearing these *PDGFRB* fusions and allowed for a precise definition of the breakpoint localization, which was then confirmed using a targeted NGS gene panel [47].

Two recent studies carried out clinical validation of OGM for detecting cytogenomic aberrations in hematological malignancies, precisely defining the OGM technical and analytical performance [46,48]. These studies both demonstrated that OGM is characterized by high sensitivity, specificity, and accuracy; OGM is highly concordant with standard cytogenetic techniques but is able to detect novel, clinically significant SVs not detected by other methods in a high percentage of examined cases, therefore allowing for better patient classification, prognostic stratification, and therapeutic choices [48]. Thus, OGM can now be considered a highly reliable method used as a first-tier cytogenomic test in hematological malignancies [46,48]. Several OGM studies have been carried out on specific MDS, AML, ALL, CLL, and MM patients (Table 2).

### 4.1. Acute Myeloid Leukemia (AML) and Myelodysplastic Syndrome (MDS)

Acute myeloid leukemia (AML) and myelodysplastic syndromes (MDSs) are common hematological malignancies in adults, characterized by several recurring mutations and cytogenetic abnormalities [59]. Therefore, diagnostics and risk stratification of AML/MDS patients are based essentially on genetic and molecular methods, such as conventional karyotyping, FISH, CNV microarrays, multiplex PCR, and NGS [59]. The use of OGM was investigated in some cohorts of patients with MDS/AML by different authors, to compare OGM results with data derived from the standard diagnostic genomic workup and to establish whether OGM can provide additional information with a potential impact on treatment [49]. Gerding et al. were the first to investigate the applicability of OGM as a diagnostic tool in a sub-group of 27 AML and MDS cases with excessive blasts [49]. Although a high (25 of 27 patients, 93%) concordance with other cytogenetic methods was observed, in several cases the redefinition of the karyotype via OGM led to the detection of cytogenetically cryptic rearrangements (*NUP98::NSD1* and *MECOM::MSI2*), the resolution of complex chromosomal aberrations including chromothripsis, the assessment of the origin of unknown marker chromosomes, and sometimes a better European Leukemia Net risk classification [59]. Only in two cases was an additional chromosome 21 missed via OGM, likely present at a level below the OGM detection limit.

Moreover, OGM analysis detected rare SV and CNV mapping near a sub-set of selected myeloid genes (such as *KMT2A-PTD*, *ETV6*, *JAK2*, and *RUNX1*) and with potential clinical relevance. Recently, in a French series of 68 adult MDS and AML patients (27 MDS and 41 AML cases), OGM methodology, that can detect most (85%) cytogenetic abnormalities observed via conventional cytogenetics, allowed for the identification of partner genes of driver genes involved in balanced rearrangements and the analysis of cytogenetic abnormalities in cases of karyotype failure or with poor quality material [33]. In addition, OGM analysis provided a complete definition of complex rearrangements and detected recurrent events involving chromosomes 12 and 21 in several cases with a complex karyotype. In fact, *ETV6* and *ERG* gene deletion and amplification, respectively, were identified in 6 of 13 cases with a complex karyotype. Moreover, the presence of novel cytogenetic abnormalities was observed in 33% and 54% of MDS and AML patients with normal routine cytogenetic data, respectively. OGM analyses unveiled recurrent alterations involving novel genes of interest in AML, such as the *KMT2A* gene partial tandem duplication (7/41 AML cases), alterations of the *MYB* gene (3/41 AML cases), and rearrangement of the *NUP98* gene (2/41 AML cases). The limitations of the OGM technique include clones at low frequency and SVs in poorly covered regions such as centromeric and telomeric regions; these circumstances explain why, in a low percentage of cases, OGM is not able to reveal cytogenetic abnormalities seen via routine cytogenetics [33].

The performance of OGM was evaluated by Levy et al. in a cohort of 100 AML cases, previously analyzed using karyotype alone or karyotype, FISH, and/or CMA[50]. Although OGM detected all of the SVs and CNVs found via cytogenetics at allelic fractions ≥ 5%, a more accurate breakpoint definition and the identification of novel clinically relevant SVs/CNVs were achieved in 13% of cases. Among cryptic translocations, the *NSD1::NUP98* and *ETV6::MECOM* fusion genes were identified via OGM analysis in 6% of cases with a normal karyotype. In addition, OGM results could modify the recommended clinical treatment (4% of cases) or identify patients eligible for enrolment in clinical trials (8% of cases). On the other hand, 33% of cases showed discrepancies between FISH and OGM, as the latter analysis is not a suitable test for identifying genetic abnormalities with low-level frequency. Since the VAF on the total DNA used for OGM is not affected by cultural artifacts, it is possible that variants missed with OGM (with initial VAF < 0.5%) may be overestimated during the cell culture process.

Suttorp et al. performed the first comparative study between OGM and classical cytogenetics in 24 pediatric patients with AML, bi-lineage leukemia, and mixed-phenotype acute leukemia [51]. In a significant proportion of cases (70%), discrepant results were obtained. Due to the higher resolution and whole-genome approach of OGM, new alterations were identified and validated for use as patient-specific markers for minimal residual disease: a deletion at chromosome 19 involving the gene *MEF2B* and two translocations, t(2;12) and t(8;12), affecting the ETV6 gene. However, OGM presents some pitfalls in identifying aneuploidies in sub-clones and translocations or CNVs in repeated regions and on the p-arm of acrocentric chromosomes. Moreover, special attention should be paid to CNV filter settings at low blast counts. Yang et al. performed high-resolution structural variant profiling in a cohort of 101 new MDS patients using the OGM technique and investigated the prognostic and clinical impact via simultaneous targeted NGS-based mutational analysis [39].

Several cryptic SVs (such as *MECOMr*, *NUP98r*, *KMT2Ar*, and *KMT2A-PTD*) were revealed in 34% of patients with a normal karyotype. Furthermore, risk stratification in MDS was improved using OGM, inducing changes in the comprehensive cytogenetic scoring system (CCSS) and the R-IPSS risk groups in 21% and 17% of patients, respectively. Moreover, in 13% of cases, a targetable alteration, useful for prognosis/therapy, was detected via OGM without changing the CCSS/R-IPSS score.

Finally, all of these data showed that OGM is a promising technology for integration with targeted NGS as part of the diagnostic MDS/AML workup.

### 4.2. Acute Lymphoblastic Leukemia

ALL is a heterogeneous hematopoietic malignancy characterized by accumulation in the blood and bone marrow of lymphoid progenitor cells belonging to the B lineage (85%) or, less frequently (15%), the T lineage. Defining chromosome ploidy and identifying specific chromosomal or molecular rearrangements in ALL is crucial to determine prognosis and choose the most appropriate treatment [60,61]. ALL cases are generally characterized by a high number of genomic alterations at diagnosis, and their definition is complex, costly, time-consuming, and requires the use of different complementary techniques such as karyotyping, FISH, SNP-array, RNA-Seq, and MLPA. Several recurrent categories of chromosomal anomalies influencing the generation of different fusion genes have been identified in B-ALL, but many of them are not detectable via conventional cytogenetics, as they are cryptic or below the limit of resolution [25]. Therefore, new technologies that can simplify and reduce the turnaround time for diagnosis and prognostic stratification are urgently needed in ALL. To date, different groups have investigated whether OGM analysis can effectively replace current standard tests employed for the genomic and molecular characterization of ALL cases at diagnosis. All of these studies have shown that the employment of OGM avoids the need for all of the standard tests generally used at diagnosis to perform the molecular and cytogenetic characterization of ALL patients, reducing turnaround times and increasing the detection resolution of genomic rearrangements.

The first preliminary data using OGM in ALL were reported by Lestringant et al. [42]; they analyzed 10 B or T-ALL patients, comparing OGM with results obtained from previous analysis via standard techniques. Concordant results were obtained in 90% of genomic abnormalities detected using both standard techniques and OGM; twelve new alterations were revealed only via OGM, whereas eight alterations were identified only via traditional methods [42]. OGM revealed four of the eight discrepancies in raw data analysis after removing filtering, as they showed low allelic frequency; the remaining four alterations were not identified, probably because of poor coverage in specific genomic regions. Among the alterations not identified, two were CNVs involving the *CRLF2* gene in the pseudoautosomal region *PAR1* in Xp22.33, which is not correctly detected via OGM [42]. On the other hand, seven out of twelve alterations detected via OGM were confirmed and validated using other techniques. Among these alterations were new inter-chromosomal rearrangements leading to some fusions such as *LMO2::TCRA*, *TCRB::MYC*, and *IGH::CEBPB* [42].

A second study by Lühmann et al. showed that OGM allowed for the detection of all genomic alterations present in ALL (translocations, aneuploidies, and copy number variation) in retrospective and prospective study samples [41]. They demonstrated that OGM allows for the detection of all structural and numeric genomic alterations identified using standard diagnostic tests in a single experiment, simplifying the diagnostic workflow. It can also identify new structural variants useful for better defining prognosis and therapeutic choice. In this study, twelve pediatric ALL samples were analyzed using OGM, and all genomic aberrations such as translocations, aneuploidies, and copy number variations identified using standard diagnostic techniques were confirmed via OGM. Moreover, OGM better defined a new rearrangement, t(9;11)(24.1;q22.1), generating a fusion involving *JAK2* and *NPAT* genes and a complex three-way translocation, t(2;12;21)(p22.1;p13.2;q22.12), producing the *ETV6::RUNX1* fusion. Moreover, OGM showed higher sensitivity for detecting copy number variations [41].

Work by Rack et al. applied OGM to the diagnostic workup of 41 ALL cases (29 B-ALL and 12 T-ALL), mainly retrospectively studied, while only three patients were prospectively analyzed [40]. The OGM results obtained in these last patients were subsequently compared to those achieved using standard diagnostic tests. Filter setting was first defined by analyzing the 38 retrospective cases, and then the same filtering was applied to the 3 prospective cases that were studied blind. Overall, all of the alterations identified using standard techniques were confirmed via OGM, but additional rearrangements generating fusion genes were revealed, allowing for the reclassification of 3 patients as B-ALL with *TCF3::PBX1* and *BCR::ABL1*-like B-ALL. Only two cases were misinterpreted: in the first patient, a large structural rearrangement involving chromosomes 2 and 14 was detected via CBA in about 90% of the analyzed cells, but it was not confirmed via OGM, probably as a consequence of breakpoint localization in repetitive regions [40]. In the second case showing discrepancy, OGM identified a fusion that was not confirmed via FISH nor RNA-Seq, probably because the same labeling fluorescent pattern was detected in the two involved regions [40]. An interesting recent application of OGM was the molecular and genomic characterization of isolated cell populations in a study investigating the occurrence of relapse in B-ALL cases treated with CD19-directed immunotherapies with CAR-T cells [52]. OGM analysis was performed on a specific CD34 + CD19-CD22+ early cell population present at diagnosis of B-ALL patients and probably responsible for relapse after CD19-targeted immunotherapy. The analysis was used to investigate whether this early specific cell population was characterized by the same molecular and genomic alterations detected at the disease onset and, therefore, could be an early leukemic clone existing before CAR T-cell therapy [52]. Another specific application of OGM in B-ALL performed by Jean et al. investigated intragenic tandem multiplication of *PAX5* (*PAX5-ITM*) in pediatric B-ALL. *PAX5* alterations are detected in about 30% of B-ALL cases, and *PAX5-ITM* is a rare mutation that consists of additional copies of different exons at the 5′ end of the gene [53]. OGM was used to determine the amplified regions of a specific copy number and define their intragenic position and orientation. In this study, OGM showed a better resolution than CMA in defining the exact copy number of amplified regions, whereas CMA analysis significantly underestimated the number of *PAX5-ITM* repetitions [53]. Finally, a recent study combined OGM technology with single-cell targeted DNA sequencing in 12 B-ALL samples to investigate clonal heterogeneity at diagnosis and during treatment [54]. OGM was employed with other techniques such as FISH, MLPA, karyotyping, and RNA-Seq to investigate the structural variants present in B-ALL patients. The data from these analyses were combined with single-cell DNA sequencing to examine the co-occurrence of sub-clonal mutations and to highlight clonal evolution during chemotherapy [54].

### 4.3. Chronic Lymphocytic Leukemia

Chronic lymphocytic leukemia (CLL) is a common lymphoproliferative disorder caused by mature CD5+ B cell clonal expansion in the blood and lymphoid tissues. CLL patients show heterogeneous biological features and variable clinical manifestations, ranging from indolent disease to rapid progression [62]. This heterogeneity can be explained by a complex landscape of genomic alterations consisting of chromosomal imbalances such as del(13q), del(11q), del(17p), and trisomy 12, which are present in about 80% of CLL patients, and a heterogeneous pattern of mutated genes (e.g., *SF3B1*, *NOTCH1*, and *ATM*) that can be detected in 10–20% of cases [63,64]. Many prognostic and predictive genomic alterations, such as 17p13 deletion, *TP53* gene mutation, complex karyotypes, and the mutational status of the *IGHV* gene, have been identified over the years [62,65]. Two large studies showed that genomic complexity in CLL is an independent poor prognostic factor; indeed, a cut-off of at least five aberrations in the same clone was defined as a predictor of worse evolution [66,67]. However, cytogenetic and molecular techniques such as CBA, FISH, and mutational analysis currently employed to assess CLL genomic alterations could underestimate the true genomic complexity of this neoplasm. A study by Puiggros et al. compared the OGM performance in defining genomic alterations of CLL patients with data obtained from standard cytogenetic techniques, and focused on detecting genomic complexity and its use as a prognostic factor [55]. Forty-two CLL patients previously analyzed using CBA, FISH, and CMA were included. Globally, 90% of the previously detected alterations were confirmed via OGM, whereas 30 genomic anomalies among 309 were missed via OGM, mainly because of breakpoint localizations in the centromeric/telomeric region or because of sub-clonal occurrence.

Regarding the definition of genomic complexity, a cut-off of >10 aberrations identified a complex OGM group showing a high frequency of *TP53* aberrations and a significantly shorter time to first treatment. Overall, this new technology allowed for a better definition of genomic breakpoint localization and SV extension. The authors, therefore, concluded that OGM is a robust, helpful method to better define CLL diagnosis and manage prognostic stratification or treatment choices [55]. A recent application of OGM in CLL was carried out by Ramos-Campoy et al and analyzed nine cases with chromothripsis [36,56]. Chromothripsis has been reported in 1–3% of CLL cases, but CBA cannot detect it due to its limited resolution, and the mechanism at the basis of this complex phenomenon is still unknown. Ramos-Campoy et al. compared chromothripsis patterns detected using microarray analyses and OGM, and revealed a high concordance (88%) in rearrangement sizes and breakpoint locations. Moreover, OGM showed the occurrence of intra- and inter-chromosomal translocations involving chromothriptic regions not detected via CMA, and in one patient revealed a new catastrophic phenomenon known as chromoplexy [35,56]. The authors concluded that OGM is a novel technology providing a more detailed characterization of these complex genomic events than microarray analysis, which could help to define the mechanisms based on their occurrence [56].

### 4.4. Multiple Myeloma

MM is a heterogeneous hematological malignancy caused by plasma cell clonal expansion, in which patients show variable disease progression and response to therapy. Several genomic aberrations characterize MM, such as translocations, mainly involving the *IGH* locus on 14q32, aneuploidies, complex genomic rearrangements, copy number changes, and SNVs. Known high-risk prognostic factors in MM include the translocations t(4;14), t(14;16), or t(14;20), the deletion of the *TP53* gene at 17p13, and the gain of the 1q arm [68,69,70]. Complex chromosomal aberrations in MM may result in both chromothripsis and chromoplexy, which occur at a frequency of about 30% and are associated with poor outcomes and known high-risk genetic alterations, including *IGH* translocations or the biallelic inactivation of *TP53* [69,71]. MM is one of the first hematological malignancies in which OGM has been employed; structural variants were investigated in a primary MM genome from a single patient at two stages of tumor progression and drug response [57]. DNA molecules were immobilized on glass surfaces treated with specific chemicals, and collected images were processed to generate single-molecule-ordered restriction maps, called Rmaps. The results from optical mapping were integrated with data from DNA genomic large-scale sequencing, demonstrating widespread structural variations and an increase in mutational burden with tumor progression in the MM patient analyzed [57]. A recent pilot study used whole-genome optical mapping, in the Saphyr system from Bionano Genomics, to analyze the genomic architecture of CD138+ cells isolated from a few patients with extramedullary MM (EMM), an aggressive condition in which myeloma cells spread to other organ systems [58,72]. Little is yet known about the mechanisms at the basis of EMM onset, but growing evidence shows that genomic aberrations may contribute to EMM pathogenesis. In this pilot study, all of the analyzed EMM patients were compared to intramedullary MM cases and showed large (>5 Mbp) intrachromosomal rearrangements across the whole of chromosome 1, predominantly deletions and inversions encompassing hundreds of genes. This study suggested an association of chromosome 1 abnormalities with extramedullary progression in MM patients, causing the dissemination of myeloma cells from BM to other tissues [58]. Moreover, a high incidence of inter-chromosomal translocations was detected in all of the analyzed MM patients; OGM confirmed that high-risk 14q32 translocation is a common primary event in MM, but all MM patients included in the study had at least two other translocations, often involving chromosomes 2, 3, 6, and 8. Many of them led to the rearrangement of genes associated with cancer. Moreover, interchromosomal translocations were often associated with intrachromosomal rearrangements, with breakpoints in the same genomic regions [58].

## 5. Conclusions

Conventional cytogenetics has remained the gold standard for global chromosome analysis in hematological malignancies and is still an important test in diagnostic workup [73,74]. CBA, combined with ancillary techniques such as FISH and CMA, allows most of the CNVs and SVs supporting the proper clinical management of hematological patients to be identified [75,76]. However, these techniques present inherent limitations, such as a dependency on cells dividing in vitro and the resolution level, and pose some challenges in detecting alterations < 5 Mb for CBA, as well as demanding an ‘a priori’ knowledge of a specific gene or region of interest for FISH, and being ineffective in detecting SVs for CMA.

In recent years, the rapid development of whole-genome technologies that can potentially analyze all clinically significant alterations has driven the international diagnostic community to evaluate the application of these next-generation methodologies in the clinical care workup.

As many hematological neoplasms demand the detection of several clinically relevant genomic variants, which necessitate a panel of individual assays, some authors have explored the feasibility of a single, comprehensive, and affordable test that captures SNVs, CNVs, and SVs. Duncavage et al. suggested WGS as an alternative to standard analyses, emphasizing its ability to reveal CNVs and/or SVs not detected using standard methods [38]. However, WGS also presents a series of inherent clinical, logistic, technical, and financial limitations, which limit the use of this technology in most laboratories [75].

In this context, the innovative OGM analysis system may be a good option to overcome some limitations of standard techniques, without requiring an excessive economic commitment. Several recent studies performed on hematological patients have demonstrated the advantages of OGM as a useful, rapid, and unbiased whole-genome test, concordant with standard cytogenetic approaches, that can reveal and better define additional and complex genome alterations [46,48]. Most of the work based on this innovative technology was carried out on MDS, AML, and ALL, whereas little has been conducted with regard to CLL or MM, and none has been conducted on lymphomas. A possible OGM field of application could be chronic myeloid leukemia. This hematological malignancy is characterized by the recurrent translocation t(9;22) and the formation of the *BCR::ABL1* fusion gene. Our group has demonstrated that this rearrangement occurs in a fair percentage of cases, with genomic material loss in chromosomes 9 and 22 [77]. Furthermore, it has recently been observed that the remaining genomic fragments were imperfectly reassembled, generating random fusions while maintaining the leukemia-initiating *BCR::ABL1* fusion [78]. In this context, the OGM approach could unveil the presence of these fusion genes and define their biologic and prognostic meaning.

A major limitation of OGM is its inability to reveal SNVs, whose detection is mandatory for diagnosing many hematological diseases [79,80,81,82,83]. These alterations can be detected using multiple molecular approaches, but it is likely that, in the near future, OGM will be combined with whole-exome or whole-transcriptome sequencing to obtain information about all SVs, CNVs, SNVs, and/or fusion genes expressed. Such a workflow will be able to provide a complete picture of all genome alterations occurring in hematological malignancies, and so may replace the current chain of standard cytogenetic diagnostic tests.

## Figures and Tables

**Figure 1 diagnostics-13-01841-f001:**
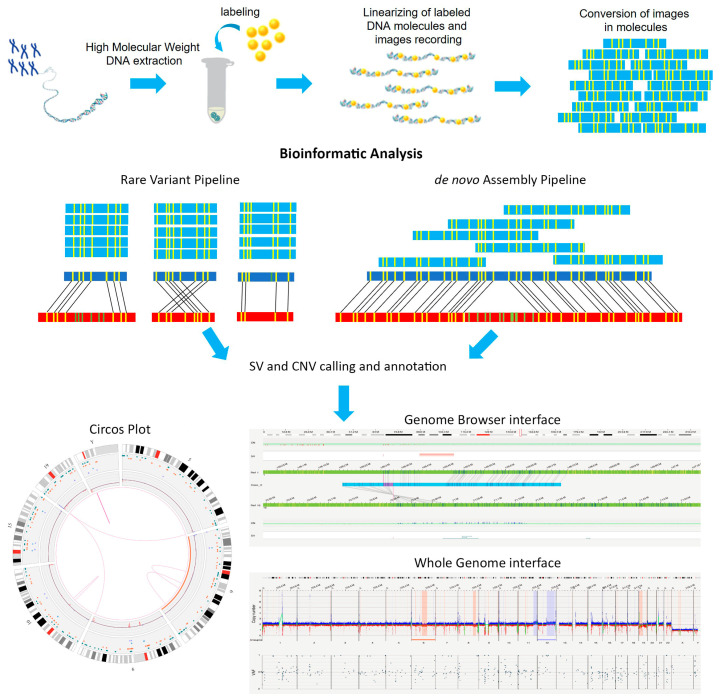
Optical genome mapping workflow. High-molecular-weight DNA isolated from the patient’s cells is labeled and loaded on a chip, linearized, and images are recorded using a Saphyr instrument. Images are then converted into molecules (sky blue) and assembled in bioinformatic pipelines. The de novo assembly arranges all of the labeled molecules in a de novo genome assembly and creates a target consensus map (blue) that is compared to the reference assembly (red). The rare variant pipeline creates clusters of molecules and compares target genomic molecule clusters (blue) to the reference genome (red). After variant calling and annotation, data can be shown using different visualization methods.

**Figure 2 diagnostics-13-01841-f002:**
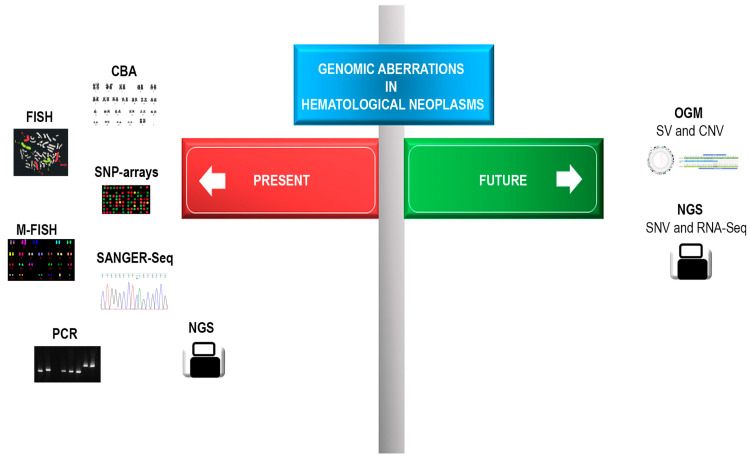
Cytogenetic and molecular diagnostic techniques used to study genomic alterations occurring in hematological malignancies. CBA: chromosome binding analysis; FISH: fluorescent in situ hybridization; M-FISH: multicolor FISH; SNP-arrays: single nucleotide polymorphism arrays; PCR: polymerase chain reaction; SANGER-Seq: Sanger sequencing; NGS: next-generation sequencing; OGM: optical genome mapping; SV: structural variations; CNV: copy number variations; SNV: single nucleotide variations; RNA-Seq: RNA sequencing.

**Table 1 diagnostics-13-01841-t001:** Advantages and limitations of OGM technique.

OGM Advantages	Limitations of Cytogenetic Standard Diagnostic Tests
Single assay instead of CBA, FISH, CMA, or MLPA	A cascade of several tests is required for a complete definition of cytogenetic alterations
Reduction in time and costs	CBA, FISH, CMA, and MLPA require more than 20 days and a few thousand dollars (USD)
High concordance with standard cytogenetic techniques	CBA often needs further confirmation
Detection of SVs (500 bp–1 Mbp) and CNVs (>0.5 Mb) at a VAF > 5–10%	CBA has a limit of about 5 Mb, FISH is based on target experiments, and CMA cannot reveal balanced alterations
Study of chromoanagenesis and better definition of complex karyotypes	Several cytogenetic tests are required to define complex rearrangements
OGM is not subject to culture bias	Both CBA and FISH require culture cells
**Limitations of OGM**	**Cytogenetic Standard Diagnostic Test Advantages**
Identification of false positive rearrangements	Some alterations identified via OGM are not confirmed via FISH or molecular analyses
False negative results	Conventional diagnostic tests are able to reveal Robertsonian translocations, telomere fusions, and isodicentric chromosomes
Inability to detect ploidy changes, CN-LOH, or single nucleotide variations (SNVs)	CBA and CMA are able to identify ploidy changes and CN-LOH
Failure to detect SVs down to a 5% allele fraction or located exclusively in centromeric/telomeric regions	CBA has the advantage of analyzing at the single-cell level and can provide information on clonal architecture

**Table 2 diagnostics-13-01841-t002:** OGM studies performed on specific series of patients with hematological malignancies.

Hematological Malignancy	No. of Patients and Type of Cancer	OGM Application	OGM Limitations	Refs.
AML, MDS	27 AML or MDS	Definition of complex rearrangements and identification of cryptic aberrations: *NUP98::NSD1, MECOM::MSI2,* and *KMT2A::PTD*	Failure to identify additional chromosome 21 because of sub-clonal occurrence	[49]
	27 MDS and 41 AML	Identification of recurrent complex rearrangements involving chromosomes 12 and 21 and novel cryptic cytogenetic abnormalities in AML: *KMT2A::PTD, MYB* alteration, and *NUP98*-rearrangement	Failure to identify some chromosomal alterations because of sub-clonal occurrence (Y loss and trisomy 8) or breakpoint localization in repetitive regions	[33]
	100 AML	Identification of novel translocations: *NSD1::NUP98* and *ETV6::MECOM*	Failure to detect sub-clonal anomalies	[50]
	24 pediatric AML	Identification of new aberrations and new minimal residual disease markers	Failure to identify aneuploidies present in sub-clones and translocations occurring in repeated regions	[51]
	101 MDS	Identification of several cryptic SVs in patients with normal karyotype and redefinition of risk-stratification	Failure to identity sub-clonal alterations	[39]
ALL	10 B-ALL or T-ALL	Identification of new fusions: *LMO2*::*TCRA*, *TCRB*::*MYC*, and *IGH*::*CEBPB*	Failure to identify four genomic anomalies because of poor coverage	[42]
	12 pediatric ALL	Better resolution of CNVs and identification of new rearrangements: t(9;11)(24.1;q22.1) and t(2;12;21)(p22.1;p13.2;q22.12)	Not reported	[41]
	29 B-ALL and 12 T-ALL	Reclassification of three patients as B-ALL with *TCF3::PBX1* and *BCR::ABL1* like	Misinterpretation of two cases because of breakpoint localization in repetitive regions	[40]
	10 CD19 CAR-treated B-ALL	Identification of genomic alterations on isolated CD34 + CD19-CD22+ cell population	Not reported	[52]
	42 pediatric B-ALL	Characterization of *PAX5*-intagenic tandem multiplications	Not reported	[53]
	12 B-ALL	Study of structural variants in combination with single-cell targeted DNA sequencing	Not reported	[54]
CLL	42 CLL	Definition of CLL genomic complexity and its use as a prognostic factor	Failure to identify 15–20% of genomic anomalies because of sub-clonal occurrence or breakpoint localization in repetitive regions	[55]
	9 CLL	Characterization of complex genomic alterations as chromothripsis and chromoplexy	Not reported	[56]
MM	1 MM	Integration between optical mapping and DNA genome sequencing to study MM progression	Use of a complex and time-consuming procedure for optical mapping	[57]
	4 EMM and 7 MM	Study of complex genomic architecture of CD138+ myeloma cells in newly diagnosed MM and EMM patients	Not reported	[58]

## Data Availability

Not applicable.

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
