# Peer review of "Feasibility of Optical Genome Mapping in Cytogenetic Diagnostics of Hematological Neoplasms: A New Way to Look at DNA"

_diagnostics, 2023, doi:10.3390/diagnostics13111841_

Round 1

Reviewer 1 Report

1.This research focused on Feasibility of Optical Genome Mapping in Cytogenetic Diagnostics of Hematological Neoplasms: a new way to look at DNA , after check the pubmed,although there were 7 related research papers, but no review papers, this manuscript was very prospective and significant.

2.It would be even better if Table 1 could clearly demonstrate the advantages and disadvantages of OGM and other methods .

3. Tables 1 and 2 were  not classic three line tables.

4. Figure1 can be much more improved.

5. If all concerns were revised, I hope this manusrcipt can be accepted, because this manuscript fully meets the requirements of this journal and  proactive and systematic.

I think English nice.

Author Response

1….”This research focused on Feasibility of Optical Genome Mapping in Cytogenetic Diagnostics of Hematological Neoplasms: a new way to look at DNA , after check the pubmed,although there were 7 related research papers, but no review papers, this manuscript was very prospective and significant”… We thank referee for this comment

2….”It would be even better if Table 1 could clearly demonstrate the advantages and disadvantages of OGM and other methods”… Table 1 has been modified and improved according to reviewer suggestions

3…”Tables 1 and 2 were  not classic three line tables”... Tables 1 and 2 have been both revised according to the Journal style

4. …”Figure1 can be much more improved”… Figure1 (now converted to Figure 2) has been much more improved enlarging fonts and images

Reviewer 2 Report

The article describes the potential optical genome mapping for identification of cytogenetic abberrations in hematological cancers. It is quite well-written manuscript, but I would like to suggest some changes. 

1. description of OGE technology on page 2 should be supported by a figure which will make it more easy to follow for readers. 

2. figure 1 should be resized and revised. All abbreviations in figure should be mentioned in caption.

3. table 1 is just summary of section 2 and 3 and is redundant. should be omitted. 

4. please ensure that there is no overlap between in the first para of section 4 and next subsections 4.1, 4.2....

5. table 2, 2nd column heading should be more specific - no. of patients and type of cancer

Minor english edits and spell checks are suggested. 

Author Response

  1. …”description of OGE technology on page 2 should be supported by a figure which will make it more easy to follow for readers”… A new figure (Figure 1) describing the OGM technology has been added as suggested.
  2. …”figure 1 should be resized and revised. All abbreviations in figure should be mentioned in caption.”… Figure 1 (now converted to Figure 2) has been much more improved enlarging fonts and images. All abbreviations are now mentioned in caption.
  3. …”table 1 is just summary of section 2 and 3 and is redundant. should be omitted”… Table 1 has been modified and improved.
  4. …”please ensure that there is no overlap between in the first para of section 4 and next subsections 4.1, 4.2”.... There is no overlap between these sections, as in the first paragraph general studies on heterogeneous series of hematological patients are discussed, whereas specific leukemia subtypes are examined in further paragraphs.
  5. …”table 2, 2nd column heading should be more specific - no. of patients and type of cancer”… Table 2 has been modified as suggested.

Reviewer 3 Report

The manuscript is well written and it is a timely topic for a review.

Minor points:

-        Page 2: consider pointing out that DNA is labeled on both genomic strands.

-        Chapter 2: "Moreover, the analysis software is easy to use and is provided with a user-friendly graphical interface that does not demand specific bioinformatics skills

-> Statement is rather subjective and overly suggestive. Other bioinformatic-heavy software solutions i. e. for NGS are currently easier to use, more reliable and user-friendly

-        Chapter 2: Consider adding FISH limitation: In rearrangements with multiple rearrangement partners (for AML: KMT2A (>100 partners), MECOM, NUP98 (>35 partners), ETV6, etc for lymphatic lineage: IGH and TRA(D) etc), break apart FISH is not able to identify fusion partner.

-> i.e. Balducci et al detected a rare NUP98::TNRC18 fusion (cited by authors, citation 46)

-        Page 3: Studies applying WGS to hematological neoplasms often set higher WGS target coverages like 60X (i.e. Duncavage et al, PMID: 33704937) for lower VAF detection limit. WGS coverage of 30X is often used for constitutional samples, therefore 60X would be more realistic.

-        Page 4 Table 1: CNVs (>0.5Mb) or (>500kb) instead of (>5Mb).

-        Page 7: The authors should consider moving Table 2 to end of chapter 4.0 prior to 4.1 (page 6). Overall, table 2 provides a great comprehensive overview over the current state of scientific research on the topic, specified in chapters 4.1-4.4.

-        Proof reading: correction of missing spaces, use of singular/plural (i. e. patients classification [patient classification] page 5, detecting smaller SV [detecting smaller SVs] page 2), and typos/placement of citations (i.e. page 12 "CNVs and SVs suppo69,70rting proper"

Author Response

- …”Page 2: consider pointing out that DNA is labeled on both genomic strands.”… This point has been commented, and the text modified (page 2, lines 49-52)

-        ..."Chapter 2: "Moreover, the analysis software is easy to use and is provided with a user-friendly graphical interface that does not demand specific bioinformatics skills

-> Statement is rather subjective and overly suggestive. Other bioinformatic-heavy software solutions i. e. for NGS are currently easier to use, more reliable and user-friendly"... The statement has been modified according to the reviewer comment and new information has been added (Page 2, lines 75-83)

-        ..."Chapter 2: Consider adding FISH limitation: In rearrangements with multiple rearrangement partners (for AML: KMT2A (>100 partners), MECOM, NUP98 (>35 partners), ETV6, etc for lymphatic lineage: IGH and TRA(D) etc), break apart FISH is not able to identify fusion partner.

 -> i.e. Balducci et al detected a rare NUP98::TNRC18 fusion (cited by authors, citation 46)"... This comment has been addressed and discussed (page 3, line 111-116)

-        …”Page 3: Studies applying WGS to hematological neoplasms often set higher WGS target coverages like 60X (i.e. Duncavage et al, PMID: 33704937) for lower VAF detection limit. WGS coverage of 30X is often used for constitutional samples, therefore 60X would be more realistic”… The referee suggestion has been accepted and the reference has been added (page 4 line 138)

-        …”Page 4 Table 1: CNVs (>0.5Mb) or (>500kb) instead of (>5Mb)”… The correction has been made

-        Page 7: …”The authors should consider moving Table 2 to end of chapter 4.0 prior to 4.1 (page 6). Overall, table 2 provides a great comprehensive overview over the current state of scientific research on the topic, specified in chapters 4.1-4.4.”… Table 2 has been moved to the end of chapter 4.0 as suggested

 Comments on the Quality of English Language

-        Proof reading: correction of missing spaces, use of singular/plural (i. e. patients classification [patient classification] page 5, detecting smaller SV [detecting smaller SVs] page 2), and typos/placement of citations (i.e. page 12 "CNVs and SVs suppo69,70rting proper" All suggested corrections have been performed

Reviewer 4 Report

In principle everything is covered and is in place. Maybe a figure explaining the principle of the detection of the SV and CNV could help to understand the mechanics of this OGM tool. There is a phrase that analysis is not difficult and easy to do. I would suggest to add a few words with which available tools and packages it can be done. Or do authors mean easy to visualize? It is not the same.

Small notes: page 2 typo "thestructural" in one word. Page 5 abbreviation RVP is not explained. In general the text is full of abbreviations. Make sure you explain all of them in one place.

Author Response

…”In principle everything is covered and is in place. Maybe a figure explaining the principle of the detection of the SV and CNV could help to understand the mechanics of this OGM tool. There is a phrase that analysis is not difficult and easy to do. I would suggest to add a few words with which available tools and packages it can be done. Or do authors mean easy to visualize? It is not the same”… ”… A new figure (Figure 1) describing the OGM technology and the main steps of bioinformatic analysis has been added. Moreover, the text has been modified as suggested (page 2, lines 75-83)

..."Small notes: page 2 typo "thestructural" in one word. Page 5 abbreviation RVP is not explained. In general the text is full of abbreviations. Make sure you explain all of them in one place".... RVP abbreviation has been introduced at page 2, line 63